# Humoral immune response to SARS-CoV-2 and endemic coronaviruses in urban and indigenous children in Colombia

Nathalie Verónica Fernández Villalobos [1,13], Patrick Marsall [2,13], Johanna Carolina Torres Páez[1], Julia Strömpl[3], Jens Gruber[2], Martín Lotto Batista[1,4], Daria Pohl[3], Gustavo Concha[5], Hagen Frickmann [6,7], Fernando Pio de la Hoz Restrepo[8], Nicole Schneiderhan-Marra [2], Gérard Krause[3,9,10,11], Alex Dulovic[2], Monika Strengert [3,11,14✉] & Simone Kann[12,14]

## Abstract

**Background** Although anti-SARS-CoV-2 humoral immune responses and epidemiology have been extensively studied, data gaps remain for certain populations such as indigenous people or children especially in low- and middle-income countries. To address this gap, we evaluated SARS-CoV-2 seroprevalence and humoral immunity towards the parental B.1 strain, local SARS-CoV-2 variants, and endemic coronaviruses in children from Colombia from March to April 2021.

**Methods** We performed a cross-sectional seroprevalence study with 80 children from Bogotá and expanded our analysis by comparing results with an independent observational study of 82 children from the Wiwa community living in the north-eastern Colombian territories. Antibody IgG titers towards SARS-CoV-2 and the endemic coronaviruses as well as ACE2 binding inhibition as a proxy for neutralization towards several SARS-CoV-2 variants were analyzed using two multiplex-based immunoassays.

**Results** While we find seroprevalence estimates of 21.3% in children from Bogotá, seroprevalence is higher with 34.1% in Wiwa children. We observe a robust induction of antibodies towards the surface-exposed spike protein, its S1-, S2- and receptor-binding-subdomains in all SARS-CoV-2 seropositive children. Only nucleocapsid-specific IgG is significantly lower in the indigenous participants. ACE2 binding inhibition is low for all SARS-CoV-2 variants examined. We observe a dominance of NL63 S1 IgG levels in urban and indigenous children which suggests an early exposure to this respiratory virus independent of living conditions and geographic location. SARS-CoV-2 seropositivity does not correlate with antibody levels towards any of the four endemic coronaviruses indicating the absence of cross-protective immunity.

**Conclusions** Overall, antibody titers, but in particular ACE2 binding inhibition are low within Colombian samples, requiring further investigation to determine any potential clinical significance.

### Plain Language Summary

Our knowledge of SARS-CoV-2, the virus causing COVID-19 remains incomplete for certain populations including indigenous people and younger age groups. Here, we aim to understand the extent to which children from urban and indigenous populations of Colombia were previously infected with SARS-CoV-2 and the related common cold coronaviruses. By measuring antibodies, protective proteins produced by the immune system, we find higher levels of previous SARS-CoV-2 infections in indigenous children of the Wiwa community (34.1%) compared to children from urbanized Bogotá (21.3%). Antibody levels towards the common cold coronaviruses were similar in SARS-CoV-2 infected and uninfected children suggesting immune responses to one coronavirus do not automatically protect against closely-related viruses. Further, we find low levels of protective immunity against SARS-CoV-2 in both populations. This finding warrants further investigation as it relates to reinfection risk and future vaccination strategies in these populations.

A full list of author affiliations appears at the end of the paper.

The novel zoonotic severe acute respiratory syndrome-related coronavirus (SARS-CoV-2), first characterized in January 2020, rapidly caused a pandemic and still represents a global challenge in particular for populations such as aged, immunocompromised, or chronically ill individuals who are at particular risk of severe COVID-19[1]. Despite the unprecedented speed at which immunological, virological, epidemiological, and clinical data was gained, there is still a lack of knowledge regarding younger people and indigenous population groups, especially in terms of immunity, disease frequency, course and severity[2]. Previous studies on SARS-CoV-2 in children have identified that while most infections are associated with mild symptoms only[3], they are correlated with high viral loads[4]. The majority of paediatric infections occur intra-familiarly[5,6], although transmission in schools and childcare settings was also described[7].

While SARS-CoV-2 epidemiology has been extensively studied in the Western Hemisphere, data from Latin America is scarce. Few population- and hospital-based studies were conducted in Colombia[8], Brazil[9,10], and Mexico[11]; equally limited is information on clinical infection characteristics in children[12–14]. Studies on SARS-CoV-2 in indigenous populations are still rarer and are often focused on seroprevalence and lethality outcomes[15].

The first PCR-confirmed SARS-CoV-2 infection in Colombia was detected on 6 March 2020, with a corresponding initial epidemic peak reached on 18 August 2020, followed by the beginning of a second wave in January 2021[16]. To mitigate the impact of SARS-CoV-2 during the first year of the pandemic, the Colombian government implemented a series of non-pharmaceutical interventions such as restricting mobility, social distancing, and closing schools, airports, and shopping centers[16]. After lifting those measures, the government again restricted international flights and implemented regional lockdowns in December 2020 and January 2021 to contain increasing caseloads during the second wave of 2021[16].

Information on SARS-CoV-2 spread across Colombia originates mostly from a population-based study performed in 10 cities between September and December 2020, which reported a seroprevalence range of 26–68%[8]. The same study estimated an unadjusted seroprevalence without considering sampling weights, test performance, or clustering level in the whole Colombian territory of 36% (95% CI: 34–39%) for children aged 5–10 years, of 38% (95% CI: 36–40%) for children aged 10–18 years, and of 51% (95% CI: 48–55%) in indigenous populations[8]. SARS-CoV-2 outcomes and data in indigenous people from Colombia have been explored in terms of seroprevalence in the department of Vaupés[17], circulating variants in the department of Amazonas[18], and nationwide mortality[19]. The latter study found a 27% higher risk of dying of COVID-19 among confirmed cases in indigenous people compared to non-indigenous Colombians mestizos[19]. The Colombian COVID-19 vaccination campaign began on 17 February 2021 and followed like other countries prioritization by occupation, age and pre-existing conditions before doses were made available to the entire population. Children from the age of three were only vaccinated from October 2021[20].

As in other countries, SARS-CoV-2 variants with distinct severity and transmission potential dominated throughout the pandemic in Colombia, where first sequencing results became available in late 2020, showing mostly the B.1 isolate[8]. From April 2021, the Mu variant was most frequently detected, which was succeeded by the Delta variant in September 2021[21]. While there is prior information on SARS-CoV-2 prevalence in Colombia, a detailed characterization of humoral immunity including neutralization capacity is lacking both towards the original SARS-CoV-2 B.1 isolate and historic and current SARS-CoV-2 variants such as Mu, Gamma, and Omicron.

By using two previously established multiplex immunoassays, MULTICOV-AB and RBDCoV-ACE2, we analyze humoral immunity towards SARS-CoV-2 and the endemic common cold coronaviruses in indigenous children from the North-East of Colombia and in urban children from its capital, Bogotá. In addition to describing seroprevalence and humoral coronavirus immunity, we also examine SARS-CoV-2 transmission routes, symptoms, and disease severity in urban children. We find a higher seroprevalence in indigenous children (34.1%) compared to urban children (21.3%), with a robust induction of antibodies against the spike protein, its S1-, S2- and receptor-binding-subdomains in all SARS-CoV-2 seropositive children, but lower levels of nucleocapsid IgGs in the Wiwa participants. ACE2 binding inhibition as proxy for neutralization is low for all SARS-CoV-2 variants examined. Among the endemic human coronaviruses (hCoVs), we identify a predominance of NL63-specific IgG S1 levels which suggests early exposure to this respiratory virus independent of living conditions and geographical location. Further, no significant differences in hCoV S1 IgG titers is present in SARS-CoV-2 seropositive or -negative individuals pointing towards the absence of cross-protective immunity. Overall, our findings highlight the need for further research to investigate variations in antibody responses among diverse population groups and to explore the clinical significance of our observations.

## Methods

**Study design and participants.** The present study was implemented to analyze immunity towards SARS-CoV-2 and the endemic human coronaviruses in children living in urban Bogotá and from the indigenous Wiwa community. The samples analyzed in this study were collected as part of two previous studies conducted in Colombia in March and April 2021[22,23]. A detailed overview on the sample selection strategy from the two initial studies is provided in Supplementary Figs. 1 and 2.

*Urban children population.* The present study is an extension to a cross-sectional survey to determine hepatitis A virus and hepatitis E virus seroprevalence in children aged 5–18 years[22]. Serum samples isolated from 5 ml of venous blood were collected from children and adolescents (further referred to as children) in Bogotá in March 2021. Inclusion criteria for the study were to live in Bogotá, to study in the selected schools, and to have the authorization and be accompanied by a parent or a legal guardian. After the emergence of SARS-CoV-2, children and their companions with acute respiratory symptoms were excluded from participation to prevent SARS-CoV-2 transmissions in the study center. All participants suffering from comorbidities associated with an increased risk of severe COVID-19 or any diseases and conditions listed in Supplementary Table 1, were also excluded from the study. All participants with an insufficient blood volume left for SARS-CoV-2 serological analysis were also excluded. In addition to the previous questionnaire, we also started to collect information on previous SARS-CoV-2 PCR results or other variables associated with a potential SARS-CoV-2 infection such as a healthcare worker in the family[24], prior travel history to regions with confirmed COVID-19 cases[25], and the use of anti-inflammatory medication such as paracetamol which are frequently used to treat symptoms of a respiratory infection (Supplementary Table 2)[26,27]. Self-reported data on COVID-19 associated symptoms within a three-month period ahead of the study participation was also documented (Supplementary Table 3). The electronic data capture tool REDCap 7.3.6, which is hosted at the Unidad de Informática y Comunicaciones - Facultad de Medicina - Universidad Nacional de Colombia[28] was used to collate data from the participant's questionnaires.

*Indigenous population*. Serum samples from individuals of the Wiwa community living in the Sierra Nevada de Santa Martha in the North-East of Colombia were collected between 28 March 2021 and 26 April 2021 as part of an independent observational study[23]. As part of this initial study, sera and nasal swabs were sampled to assess the extent of SARS-CoV-2 dissemination by using Point-of-care (POC) SARS-CoV-2 antibody-, POC antigen-, and polymerase chain reaction (PCR)-based molecular testing[23]. All study participants received their SARS-CoV-2 molecular and serological test result as part of the previous study. For our study, we selected Wiwa children (≤18 years old) and excluded Wiwa adults (>19 years old). From the 99 Wiwa children participating in the previous study, we excluded 17 individuals with a positive SARS-CoV-2 PCR/antigen test result at the time of serum sampling from our analysis.

**Ethical approval**. Both studies were performed in line with the Declaration of Helsinki. The study conducted in Bogotá was approved by the Comité De Ética De Investigación De La Facultad De Medicina, Universidad Nacional de Colombia, Bogotá, Colombia (N°.009-125-19 and N°. 011-083) and by the Ethics Committee of Hannover Medical School, Hannover, Germany (Nr.9254_BO_K_2020). The study conducted in the north-eastern Colombian territory was approved by the Ethics Committee for Science of the University Area Andina, Bogotá, Colombia (number 1304211). All participants or their legal representatives provided written informed consent prior to study start. Participation in both studies was voluntary.

### Serological assays

*MULTICOV-AB*. Semi-quantitative IgG antibody titers and binding towards variants of concern and the endemic coronaviruses were analyzed using MULTICOV-AB, a multiplex coronavirus immunoassay that utilizes an antigenic panel of 20 coronavirus proteins, including those from SARS-CoV-2 and the endemic coronaviruses OC43, HKU1, NL63, and 229E. All antigens contained in MULTICOV-AB are listed in Supplementary Table 4. The assay was performed as previously described[29]. In brief, antigens were immobilized on magnetic MagPlex beads (Luminex Corporation) by EDC/s-NHS coupling. The individual bead populations were then combined into a mastermix and incubated on 96-well plates (Corning Costar, cat no: 3355) with 1:400 diluted sera samples. Unbound antibodies were removed by washing, and IgG was then detected using R-phycoerythrin labeled goat-anti-human IgG (Jackson ImmunoResearch Labs, cat no: 109-116-098, lot: 149288). After another wash and a bead resuspension step, samples were measured once using a FLEX-MAP 3D instrument (Luminex Corporation). Quality control (QC) samples were included on each plate to control plate-to-plate variation. Data is presented as normalized values (median intensity fluorescence (MFI)/MFI of QC sample). SARS-CoV-2 seropositivity is defined as IgG Signal to Cut-off (S/CO) of ≥1.0 for both the spike trimer and receptor-binding-domain (RBD) antigen.

*RBDCoV-ACE2*. ACE2 binding inhibition towards the RBD antigen of the parental B.1 isolate, Gamma P1, Mu B.1.621, and the Omicron sub-lineage BA.1 were assessed using the surrogate neutralization assay RBDCoV-ACE2. RBDCoV-ACE2 functions as a plate-based multiplex competitive ACE2 inhibition assay[30]. Details on the assay's antigen panel are listed in Supplementary Table 5. Briefly, SARS-CoV-2 RBDs were immobilized on magnetic MagPlex beads by Anteo coupling with the AMG Activation Kit for Multiplex Microspheres (cat no: A-LMPAKMM-400,

Anteo Technologies) following the manufacturer's instruction. Beads were incubated with biotinylated ACE2 and individual 1:400 diluted samples in a 96-well plate (Corning Costar, cat no: 3686). Bound ACE2 was detected using Strep-PE (cat no: SAPE-001, Moss). Samples were measured once on a FLEXMAP 3D instrument with the same settings as MULTICOV-AB and analyzed by normalization of MFI values against the control. 100% ACE2 binding inhibition indicates maximum binding inhibition between the corresponding SARS-CoV-2 RBD and ACE2. Responders are classified above an ACE2 binding threshold of 20%, as defined in Junker et al. [30].

*SARS-CoV-2 IgG ELISA*. All samples were additionally analyzed with the Anti-SARS-CoV-2-ELISA IgG (cat no: EI 2606-9601G, EUROIMMUN) according to the manufacturer's instructions. SARS-CoV-2 seropositivity is defined as a semi-quantitative S/CO ratio of ≥1.1 towards the S1 antigen.

**Data analysis and statistics**. SARS-CoV-2 seropositivity determined by MULTICOV-AB was defined as the main outcome. For urban children, we further studied socio-economical and behavioral aspects as well as self-reported symptoms as exposure variables.

While we present the participants' main characteristics in tables, categorical variables are summarized as counts or percentages, and continuous variables as medians or inter-quartile ranges (IQR). We utilized the Fisher's exact test to identify an association between our main outcome and other categorical variables. Odds ratios (ORs) and their 95% confidence intervals (95% CI) were calculated to study any association between the main outcome and continuous variables by using the Generalized Linear Models (GLM) with binomial family, logit link, and Maximum-likelihood (ML) estimation.

We calculated the crude and an adjusted SARS-CoV-2 seroprevalence with 95% CI for both serological assays. The adjusted seroprevalence was calculated as proposed by Lang and Reiczigel[31] (Supplementary Table 6) to improve comparability between assays. The agreement between the SARS-CoV-2 ELISA and MULTICOV-AB was calculated by Fleiss's k statistic with 95% CI[32].

When comparing values of a quantitative variable between groups, we used the Wilcoxon rank sum test[33]. Spearman's ρ was used for correlation analysis between two quantitative variables[34]. Statistical significance was defined as $p < 0.05$.

Data visualization and statistical analysis were performed in R studio version 4.0.2[35]. Information about specific R add-on packages used for statistical analysis and graphical displays are listed in Supplementary Table 7. Graphs exported from R studio were further edited in Inkscape 0.92[36].

**Reporting summary**. Further information on research design is available in the Nature Portfolio Reporting Summary linked to this article.

### Results

**SARS-CoV-2 seroprevalence in urban and indigenous children from Colombia**. First, we analyzed SARS-CoV-2 seroprevalence in our study population using MULTICOV-AB (Table 1). Within the urban children population, the unadjusted seroprevalence was 21.3% (95% CI: 13.2–32.1%; $n = 80$), while in indigenous Wiwa children seroprevalence reached 34.1% (95% CI: 24.3–45.5%; $n = 82$) in March and April 2021. As additional control, we measured SARS-CoV-2 seroprevalence with a commercial IVD-certified ELISA which led to comparable levels (Table 1 and Supplementary Fig. 3). This was also reflected in a Fleiss κ

**Table 1 SARS-CoV-2 seroprevalence measured with MULTICOV-AB and the EUROIMMUNE S1 ELISA in urban and indigenous children from Colombia.**

|  | Urban children | Indigenous children |
|---|---|---|
| Sample collection | March 2021 | March–April 2021 |
| Total participants, n | 80 | 82 |
| Age (years), median (IQR) | 11 (9–14) | 12 (9–14) |
| Sex (female, n, %) | 43 (53.8) | 50 (61.0) |
| SARS-CoV-2 seroprevalence using MULTICOV-AB | | |
| Reactive participants, n | 17 | 28 |
| Crude seroprevalence, % (95% CI)[a] | 21.3 (13.2–32.1) | 34.1 (24.3–45.5) |
| Adjusted seroprevalence, % (95% CI)[b] | 24.1 (13.8–35.2) | 38.7 (26.8–50.7) |
| SARS-CoV-2 seroprevalence using SARS-CoV-2 ELISA EUROIMMUN | | |
| Reactive participants, n | 18 | 24 |
| Crude seroprevalence, % (95% CI)[a] | 22.5 (14.2–33.5) | 29.3 (20.0–40.5) |
| Adjusted seroprevalence, % (95% CI)[b] | 25.5 (12.9–38.9) | 34.3 (20.9–48.1) |

[a]Borderline ELISA results were included in the calculation as non-reactive.
[b]Seroprevalence was adjusted for the respective assay's sensitivity and specificity.

coefficient of 0.86 (95% CI: 0.78–0.95) which indicates substantial agreement between assay results. To assess the risk of false-positive results in sera from individuals exposed to a distinct non-European pathogen spectrum, we verified assay performance by measuring 47 pre-pandemic samples collected in 2018 and 2019 from both populations but found only one sample to be reactive in MULTICOV-AB and none in the ELISA (Supplementary Fig. 4).

While a systematic questionnaire was not conducted as part of the study in the Wiwa community, we recorded information on potential transmission routes, symptoms, and disease severity for children from Bogotá (Supplementary Tables 2 and 3). Most IgG-reactive cases in our urban study population belonged to children attending public schools (13/17, 76.5%), had families with an income between one and two minimum wages (14/17, 82.3%) and all (100.0%) were born in Colombia. Eight participants ($n = 17$, 47.1%) reported having contact with a family member with confirmed COVID-19 diagnosis, and seven ($n = 17$, 41.2%) reported having contact with a probable or confirmed case. Of 17 children with detectable SARS-CoV-2 IgG levels, only 6 cases were PCR-confirmed infections (35.3%). When it came to disease symptoms, the most frequent self-reported symptom by the IgG-reactive cases was nasal congestion (5/17, 29.4%), eight children ($n = 17$, 47.1%) did not report any symptoms and none of the SARS-CoV-2 IgG-reactive children classified any of their symptoms as severe.

**SARS-CoV-2 and endemic coronavirus antibody profiles in urban and indigenous children.** Next, we examined the antigenic response profiles in our study population using MULTICOV-AB[29] and found a trend towards decreased levels for the trimeric spike protein and its subdomains (RBD, S1, S2) for indigenous children compared to urban children (Fig. 1a–e), although a significant difference was only reached for the nucleocapsid antigen (Fig. 1e). We then expanded our analysis to the endemic human coronaviruses (hCoV) to determine a correlation between hCoV antibody titers and SARS-CoV-2 IgG seropositivity. When comparing SARS-CoV-2 IgG-reactive to non-reactive participants, we found a significant difference only for hCoV 229E S1 IgG titers in our children population. However, SARS-CoV-2 IgG-reactive urban children had lower 229E S1 IgG titers and

indigenous children had higher (Fig. 2d). Within the hCoVs, NL63 showed highest levels of S1-specific IgG with comparable degrees of exposure between the two populations (Fig. 2c and Supplementary Fig. 5). In contrast, S1 IgG levels for HKU1 (Fig. 2b) and 229E (Fig. 2d) were significantly higher in indigenous children with $p$-values of 0.036 and 0.0022 indicating towards an increased circulation of those viruses in the indigenous population.

**Analysis of ACE2 binding inhibition as proxy for neutralization in urban and indigenous children.** Last, we assessed ACE2 binding inhibition as a proxy for neutralization. While we had identified a robust induction of spike (subdomain)-specific antibody titers and substantial seroprevalence levels (Fig. 1 and Table 1), ACE2 responder rates were low with only 23.5% in the urban and 25.0% in the indigenous children among the seropositive participants (Fig. 3 and Supplementary Table 8). ACE2 responder rates towards the locally dominating SARS-CoV-2 variants Gamma and Mu around the time of sample collection and the more recent BA.1 were even lower or non-existent (Supplementary Table 8).

**Discussion**

We compared SARS-CoV-2 seroprevalence and humoral responses towards hCoVs and SARS-CoV-2 in children from urban Bogotá and of the Wiwa community living in the north-eastern Colombian territories. Our unadjusted seroprevalence for urban children of 21.3% (95% CI: 13.2–32.1%) in March 2021 is in line with other studies performed in the Americas during similar timeframes[8,37,38]. For instance, the population-based study conducted in Colombia by Mercado-Reyes et al.[8] estimated an unadjusted seroprevalence of 36% (95% CI: 34–39%) in the age group of 5–10 years and of 38% (95% CI: 36–40%) in the age group 10–18 years in the time period September–December 2020. For indigenous populations in Colombia, Mercado-Reyes et al. described a seroprevalence of 51% (95% CI: 48–55%) in adults[8]. However, a systematic review in indigenous populations across South America identified seroprevalence ranges between 4.2% and 81.65%[15]. In contrast to the drastic differences in SARS-CoV-2 serostatus between those indigenous populations, our seroprevalence estimates of 21.3% and 34.1% in our children population are somewhat comparable despite the differences in living conditions and the absence of non-pharmaceutical interventions in the Wiwa community.

To the best of our knowledge, this is the first study examining in detail coronavirus humoral immune responses including neutralizing activity in children from South America. We observed substantial levels of SARS-CoV-2 seropositivity; however, both SARS-CoV-2 antibody titers and the corresponding levels of ACE2 binding inhibition were surprisingly low across our study population independent of the SARS-CoV-2 isolates tested. While a low inhibition towards more "modern" variants could be expected, as primary infection occurred with another variant[39], the low overall humoral responses require further investigation to determine whether this a country/continent-wide phenomenon and/or related to assay cross-reactivity and performance as seen for other pathogens such as tuberculosis[40,41]. For further clarification and to enable a direct comparison, only Pontes et al. are among the few South American studies utilizing the Anti-SARS-CoV-2 IgG ELISA that present S1 IgG titer ratios[42–45]. Interestingly, our mean IgG S1 ratios for SARS-CoV-2 seropositive individuals are within a comparable range pointing towards a continent-wide phenomenon. In addition, previous analysis of paediatric sera samples found higher antibody titers[46,47] and ACE2 binding inhibition[46] showing the validity of

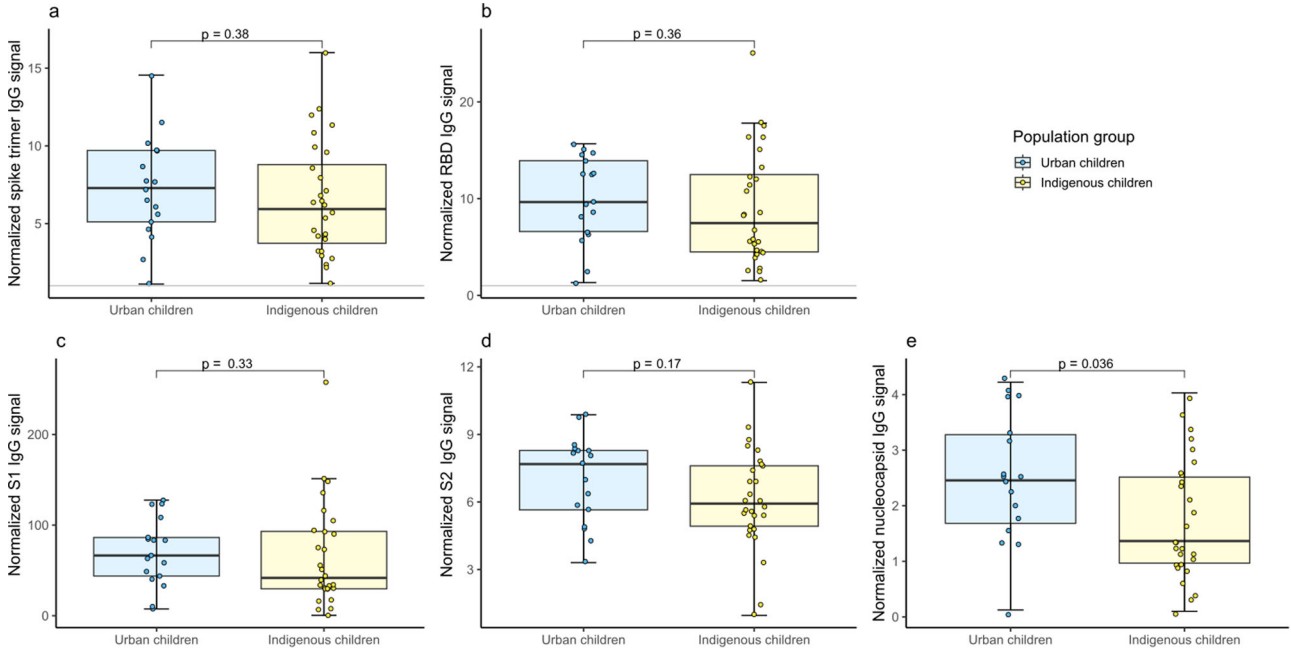

**Fig. 1 Antigenic characterization of SARS-CoV-2 antibody responses in urban and indigenous children.** IgG responses of sera samples collected in March and April 2021 were measured using MULTICOV-AB towards the spike B.1 (**a**), the RBD B.1 (**b**), the S1 B.1 subdomain (**c**), the S2 B.1 subdomain (**d**), and the nucleocapsid B.1 antigen (**e**) of SARS-CoV-2. Data is expressed as normalized MFI ratio for seropositive urban children ($n = 17$, blue) and seropositive indigenous children ($n = 28$, yellow). Seropositivity is defined as a dual cut-off of spike and RBD S/CO $\geq$ 1.0. Boxes represent the median and the 25th and 75th percentiles. Whiskers show the largest and smallest non-outliers values. Outliers were identified using upper/lower quartiles 1.5 ± times IQR. Means of two population groups were compared with the Wilcoxon's ranked sum test. Statistical significance was defined as $p < 0.05$.

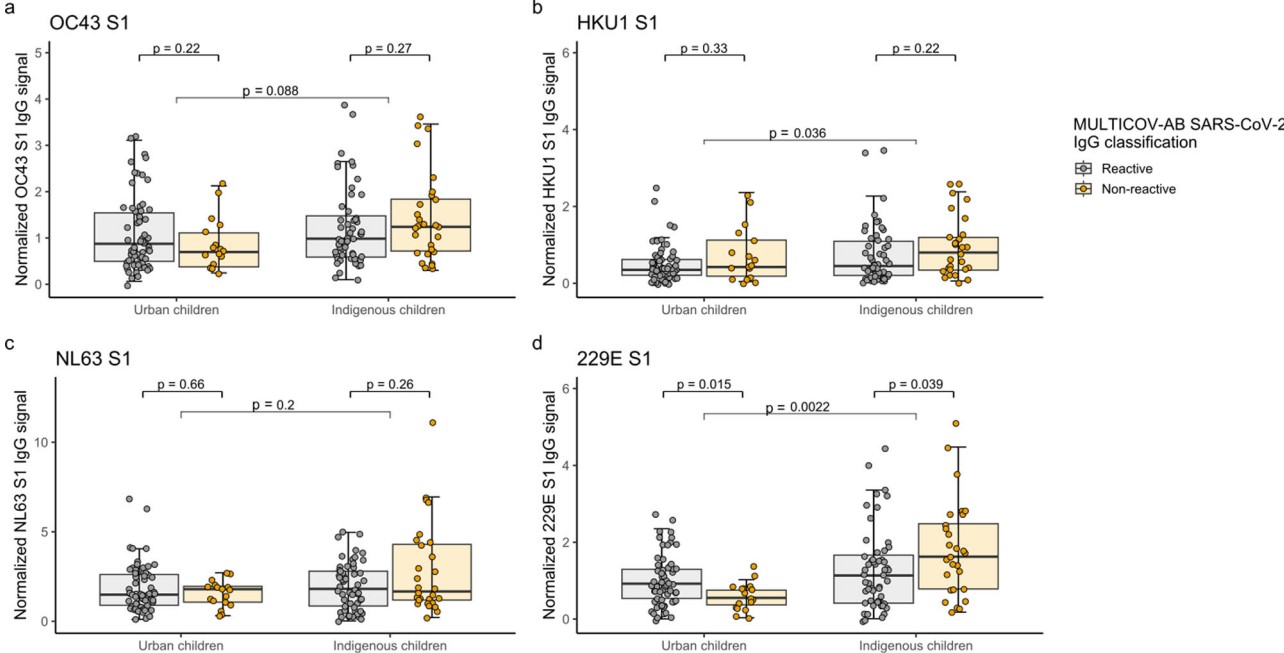

**Fig. 2 Humoral immune response towards endemic coronaviruses according to SARS-CoV-2 serostatus in urban and indigenous children.** Humoral IgG responses of urban ($n = 80$) and indigenous children ($n = 82$) towards the spike S1 subdomain of OC43 (**a**), HKU1 (**b**), NL63 (**c**), and 229E (**d**) are split based on SARS-CoV-2 serostatus. Box and whisker plots represent the median, 25th and 75th percentiles. Whiskers show the largest and smallest non-outliers values. Outliers were identified using upper/lower quartile 1.5 ± times IQR. Statistical significance was defined as $p < 0.05$ using the Wilcoxon's ranked sum test.

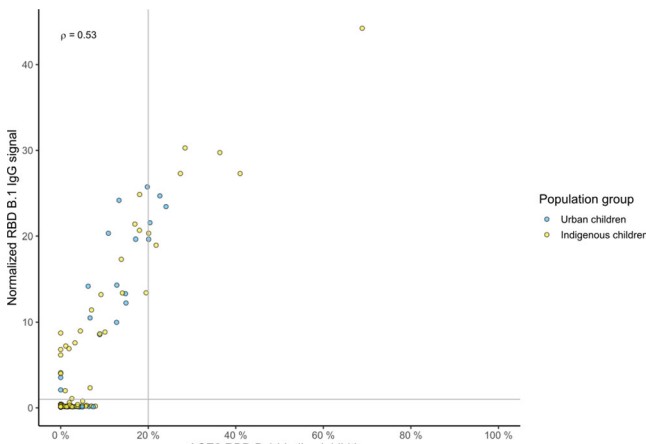

**Fig. 3 Correlation of IgG binding and ACE2 binding inhibition for the SARS-CoV-2 B.1 RBD.** Correlation analysis of ACE2 binding inhibition and IgG binding from sera samples of children from urban Bogotá ($n = 80$, blue) and from the indigenous Wiwa ($n = 82$, yellow) was performed for the RBD of the B.1 isolate. Correlation analysis was performed across all samples. The resulting Spearman's coefficient is shown in the upper left quadrant. Dashed lines indicate the ACE2 responder threshold of 20% and the S/CO of 1 for the RBD B.1 antigen.

our assay systems. In one study, sera samples from a comparable timeframe (2020-2021) were analyzed from children exhibiting mild or asymptomatic infections as part of a household transmission study[47], while another study included a broad sample selection from infected individuals (among them 20 paediatric samples with PCR-confirmed infection from the first epidemic wave in Germany) as well as from vaccinated individuals[46].

We further observed differences in antigenic response profiles between our two populations. Becker et al.[29] described a balanced induction of spike (subdomain)- and nucleocapsid-specific IgG in a group of 286 convalescent SARS-CoV-2-infected adults from Germany. In our study, this was only the case for urban children where one individual had a nucleocapsid IgG S/CO < 1. In contrast, 32.1% of Wiwa children had a nucleocapsid IgG S/CO < 1 which is the cut-off used to distinguish vaccination from infection-induced antibody responses in MULTICOV-AB[48,49]. While the functional role of nucleocapsid-specific antibodies remains to be clearly defined[39], they have been linked to COVID-19 disease severity in adults[50,51]. However, an earlier waning compared to spike-specific IgG has also been described[52–55]. This discrepancy in antigenic response profiles implies differences in response patterns might exist between different populations and stresses the limitations when using single-analyte technologies for analysis of humoral immunity[56,57]. However, neutralization responses were comparable between urban and indigenous children, which equally needs further investigation. In particular, when considering the different living conditions, hygienic standards, or access to healthcare facilities between the Wiwa community inhabiting remote areas in the Sierra Nevada de Santa Martha and urban children from Colombia's capital.

Our study has several limitations. First, we cannot conclusively determine the infection time point which might also explain the low response levels and the lack of nucleocapsid IgG, if sampling occurred late after the SARS-CoV-2 infection. Although we confirm findings of others[47,58–60] that the majority of SARS-CoV-2 infection in children is mild or asymptomatic, we cannot extend this finding to the Wiwa children as the initial study did not contain a systemic questionnaire. Last, our study did not have a longitudinal component to analyze persistence of antibody levels

or ACE2 binding inhibition nor T-cell responses. Both would have been particularly valuable considering the low levels of ACE2 binding responses identified in our study. We acknowledge that our study is primarily descriptive in nature, nevertheless, it holds significant value in increasing our understanding of the impact of COVID-19 on diverse communities. By focusing on children from both urban and indigenous populations, our research provides not only evidence on the transmission dynamics of the virus in densely populated urban areas as well as remote indigenous communities but also describes antibody responses in more detail than other studies resulting in the noteworthy finding of low neutralization and antibody levels. This finding is particularly significant as it underscores the variability in humoral immune responses to SARS-CoV-2, suggesting the need to adjust serological tools when comparing different ethnic populations or differences in the immune response exist per se. Additionally, our study examines humoral immunity towards the four endemic human coronaviruses. Our insights fill gaps in existing knowledge, especially in countries like Colombia, where such areas have been inadequately studied and can act as basis for future more mechanistic work.

Overall, we found a high proportion of individuals had already been exposed to SARS-CoV-2 in Colombia by mid-2021. Despite different circumstances of living, we further identified only minimal differences in antibody responses between the urban and indigenous populations, although immunity was reduced for both groups compared to European populations that we have previously studied after SARS-CoV-2 infections using our multiplex-based analysis approach[30,46,47]. This unexpected finding warrants further investigation into potential differences in antibody responses between separate populations as it could have implications for reinfections and vaccination responses on an individual level and impact predictions for herd immunity on a population level.

## Data availability

The datasets and data analysis supporting the conclusions of this article have been provided by the authors in a public repository (https://zenodo.org/badge/latestdoi/559828775)[61].

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

## Acknowledgements

We thank the laboratory biologists (Nicolás Lalinde Ruiz, Manuela Llano León, and Laura Camila Martínez Enríquez), nurses (Lina María Beltrán Toca, Carlos David Zabala Vega, Oscar Stiven Alméziga Clavijo, and Martha Yelitza Rodríguez Malagón), and medical personnel (Rosanna Camerano Ruiz, Karem Johanna Delgado Garcia, Andrés Felipe Mora Salamanca, and Laura Johanna Piña Jaramillo) at Universidad Nacional (UNAL) for their assistance with data collection and processing. We thank the UNAL administrative personal for their support in study planning and imple-mentation. We thank the HZI team that supported administrative and laboratory

processes. We particularly want to thank Barbora Kessel for her continued advice on data analysis. Last, we thank Philipp D Kaiser, Bjoern Traenkle, Ulrich Rothbauer, Matthias Becker, Daniel Junker, and Johanna Griesbaum for their help in producing assay components. The research was funded by intramural funds of the HZI, by the State Ministry of Lower Saxony for Science and Culture (grant agreement 14-76103-1841, MWK HZI COVID19), by the Networking Fund of the Helmholtz Association of German Research Centres (grant number SO-96) and the State Ministry of Baden-Württemberg for Economic Affairs, Labour and Tourism (grant agreement 3-4332.62-NMI/68). The indigenous study was funded by the Medical Mission Institute, Würzburg, Germany. Nathalie Fernández receives a scholarship by Studienstiftung des deutschen Volkes.

## Author contributions

G.K., F.P.H.R., N.V.F.V., M.S., S.K., A.D. and J.C.T.P. designed the study. S.K., G.C., N.V.F.V. and J.C.T.P. executed the study and collected data. G.K., M.S. and S.K. supervised the study. P.M., J.G., J.S., and T.W. performed the laboratory experiments. M.S., A.D., and P.M. supervised and coordinated laboratory work and analyzed laboratory data. N.V.F.V. cleaned the database and prepared data for analysis. A.D., M.S., P.M., and N.V.F.V. analyzed and interpreted the data. N.V.F.V. and M.S. prepared the initial manuscript. M.S., D.P., and M.L. verified the underlying data. M.S., H.F., S.K., and A.D. provided advice on data analysis. S.K. and G.C. coordinated and collected indigenous data and samples. G.K., S.K., M.S., and N.S.M. acquired funding. All authors have revised the manuscript, read, and approved the final version. All authors confirm full access to all the data in the study and accept responsibility to submit for publication.

## Funding

## Competing interests

N.S.M. was a speaker at Luminex user meetings in the past. A.D. has previously given sponsored talks for Sino Biological and Luminex. The Natural and Medical Sciences Institute at the University of Tübingen is involved in applied research projects as a fee for services with the Luminex Corporation. The other authors declare no competing interests.

## Additional information

[1]Department of Epidemiology, PhD Programme, Helmholtz Centre for Infection Research (HZI), Braunschweig-Hannover, Germany. [2]Multiplex Immunoassays, NMI Natural and Medical Sciences Institute at the University of Tübingen (NMI), Reutlingen, Germany. [3]Department of Epidemiology, Helmholtz Centre for Infection Research (HZI), Braunschweig, Germany. [4]Global Health Resilience, Barcelona Supercomputing Center (BSC), Barcelona, Spain. [5]Organization Wiwa Yugumaiun Bunkauanarrua Tayrona (OWYBT), Department Health Advocacy, Valledupar, Colombia. [6]Department of Microbiology and Hospital Hygiene, Bundeswehr Hospital Hamburg, Hamburg, Germany. [7]Institute for Medical Microbiology, Virology and Hygiene, University Medicine Rostock, Rostock, Germany. [8]Universidad Nacional de Colombia, Facultad de Medicina, Departamento de Salud Pública, Bogotá, Colombia. [9]Hannover Medical School, Hannover, Germany. [10]German Centre for Infection Research (DZIF), Braunschweig-Hannover, Germany. [11]TWINCORE, Centre for Experimental and Clinical Infection Research, a joint venture of the Hannover Medical School and the Helmholtz Centre for Infection Research, Hannover, Germany. [12]Medical Mission Institute, Würzburg, Germany. [13]These authors contributed equally: Nathalie Verónica Fernández Villalobos, Patrick Marsall. [14]These authors jointly supervised this work: Monika Strengert, Simone Kann. ✉email: Monika.Strengert@helmholtz-hzi.de

