## [Peer Review File · Communications Medicine]

Reviewers' comments:

Reviewer #1 (Remarks to the Author):

The authors conducted a seroprevalence study on SARS-CoV-2 in children in Colombia. One of the main assets of this manuscript is the target population, which included indigenous children, and also evaluating endemic coronaviruses, followed by different humoral features. The authors should be commended by the work, but this reviewer has some concerns, as below. Many thanks for the opportunity to read your work.

Major

1 - A key feature to interpret and analyse seroprevalence studies is the sampling strategy. It is not clear in the methods how this was actually done, how many were excluded, "response rate", if at the end it was a convenient sample. Please, clarify and adjust where appropriated, including the discussion of potential biases.

2 - Please, define in the methods what was considered a "seropositive" case (single Ig test, combination of different test, how was managed discordance between tests)

3 - There are several statements in results that should go in the methods section. Please, cite then in methods (lines 219? and certainly 222-224, among others that repeat methods)

4 - When evaluating the correlations, how were managed samples with undetectable or below the lower limit of detection? From Fig 3 it is not clear whether those value outside the thresholds were/how used to the Spearman's correlation.

5 - When discussing seroprevalence ranges from other studies, please, state if they were population-based (sampling schemes) and use confidence intervals. This will help the readers.

6 - This reviewer is not sure the authors have data to support statements as "immunity was reduced for both groups compared to European populations that we have previously studied". How comparable were the sampling? Was there a robust evidence of difference? Please, revise and justify.

7 - Following item 7, this statement in the abstract "in particular ACE2 binding inhibition were lower than expected within Colombian samples" is not clear to this reviewer.

Overall, the manuscript would benefit from a text review assuring a narrative flow. The current format, particularly of results, is hard to follow.

Minor

1 - Could the authors please expand the sentence about "symptomatic infections are rare" (Lines 81/82). In which context? Compared with what? General, vulnerable population? Do we have accurate data for this statement?

2 - "unadjusted seroprevalence" (Line 98), in terms of test performance? sampling weights?

3 - Is this correct "Eight participants (47.1%)"? which denominator...

4 - Please, use sex instead of gender, unless you evaluated gender.

Reviewer #2 (Remarks to the Author):

The manuscript "Humoral immune response towards SARS-CoV-2 variants of concern and endemic coronaviruses in urban and indigenous children from Colombia" by Fernández Villalobos et al. presents data on several serological techniques to measure SARS-CoV-2 and common cold coronaviruses specific antibodies in children from Colombia belonging to both urban (n=80) and rural settings (n=82) at a single time point (March-April 2021). The study also adds a surrogate neutralizing assay to measure neutralization against different SARS-CoV-2 VoCs. The study is technically solid, and it is well presented however the main problem is the limited relevance of the information on such a small number of individuals analyzed at just one specific moment. Furthermore, time of SARS-CoV-2 infection is lacking and taking into account the small number of participants significant biases can occur such as inclusion criteria, not completely define (how many offered?) or even exclusion as the authors describe in line 151: (from 99 indigenous children) we excluded 17 individuals with a SARS-CoV-2 positive PCR/antigen test result at the time of serum sampling from our analysis. This is a high rate of what seems to be active SARS-CoV-2 infection. In this same line, the rest of the information, ie the lack of association between the levels of antibodies against SARS-CoV-2 and common cold coronaviruses along with the low level of neutralization observed is already well established or lacks adequate controls respectively. Overall, it is very difficult that the current design of study can provide any significant or informative conclusion

Helmholtz Centre for Infection Research GmbH | Department of Epidemiology
Inhoffenstraße 7 | 38124 Braunschweig | Germany

Communications Medicine

Braunschweig, 15.05.2023

Dear Reviewers,

On behalf of all the authors, I would like to resubmit our revised manuscript "**Humoral immune response towards SARS-CoV-2 variants of concern and endemic coronaviruses in urban and indigenous children from Colombia**".

While we are addressing all your comments with alterations in the main text and in a detailed point-by-point reply, please also allow me to restate after a recent literature review that our study is the first of its kind for children from Colombia. It does not only present detailed data on SARS-CoV-2 antibody responses including neutralization capacity, but also defines humoral immunity towards the four common cold coronaviruses in urban and indigenous populations. Both subgroups where information on immunity and disease frequency is still scarce.

We hope you find our revised manuscript suitable for publication. We are looking forward to hearing from you in due course.

On behalf of all authors, with kind regards

Monika Strengert

Dr. Monika Strengert
Department of Epidemiology
Tel. +49 531 6181-3103
Monika.Strengert@
helmholtz-hzi.de

**Helmholtz Centre for Infection
Research GmbH**

SCIENCE CAMPUS Braunschweig-Süd
Inhoffenstraße 7
38124 Braunschweig
Germany
Tel +49 531 6181-0
Fax +49 531 6181-2655
www.helmholtz-hzi.de

Chair of Supervisory Board:
MinDir'in Prof. Dr. Veronika von Messling
Federal Ministry of Education and
Research

Deputy:
MinDirig Rüdiger Eichel, Department
Leader, Ministry for Science and Culture
of Lower Saxony

Scientific Director:
Prof. Dr. Dr. h.c. Dirk Heinz

Administrative Director:
Christian Scherf

Registered Office:
Braunschweig

Registry Court:
Amtsgericht Braunschweig HRB 477
VAT Reg. No DE 11 48 15 244

Bank account:
Braunschweigische Landessparkasse
BIC NOLADE 2H XXX
IBAN DE56 2505 0000 0002 0588 81

St.-Nr. 13/200/24006

Reviewer #1

Remarks to the Author

The authors conducted a seroprevalence study on SARS-CoV-2 in children in Colombia. One of the main assets of this manuscript is the target population, which included indigenous children, and also evaluating endemic coronaviruses, followed by different humoral features. The authors should be commended by the work, but this reviewer has some concerns, as below. Many thanks for the opportunity to read your work.

Response: We thank you for kind commendations and hope we have addressed your concerns sufficiently and adequately.

Major

1 - A key feature to interpret and analyse seroprevalence studies is the sampling strategy. It is not clear in the methods how this was actually done, how many were excluded, "response rate", if at the end it was a convenient sample. Please, clarify and adjust where appropriated, including the discussion of potential biases.

Response: In the present study, we combined serum samples from two previous studies. We describe this in the method section alongside the criteria necessary to be included in both studies in the first place. We only included samples in the present work collected in a comparable timeframe as collection time can act as a bias. Consequentially, we also excluded indigenous children with a SARS-CoV-2 positive PCR result at the time of serum sampling as seroconversion might not have occurred yet, which clearly results in a bias. For clarification, we have now included two flow charts to describe sample selection in more detail from the preexisting sample pool as Fig S1 and S2 in the supplements section.

2 - Please, define in the methods what was considered a "seropositive" case (single Ig test, combination of different test, how was managed discordance between tests)

Response: We appreciate your comment and apologize if this information was not clear in the manuscript, we have added this information where needed to improve clarity – i.e. line 219. As primary readout, we assessed SARS-CoV-2 seropositivity and antibody binding using MULTICOV-AB and as additional control the Euroimmun S1 IgG ELISA resulting in a considerable agreement between those when calculating the Fleiss coefficient. As third independent test, we used RBDCoV-ACE2 to characterize ACE2-RBD binding inhibition of SARS-CoV-2 antibodies.

To provide a more comprehensive overview, we have listed the statements from the manuscript that define the reactivity next to each other.

- MULTICOV-AB: "SARS-CoV-2 seropositivity is defined as IgG Signal to Cut-off (S/CO) of ≥ 1.0 for both the Spike Trimer and receptor-binding-domain (RBD) antigen"
- SARS-CoV-2 IgG ELISA: "SARS-CoV-2 seropositivity is defined as a semi-quantitative S/CO ratio of ≥ 1.1 towards the S1 antigen."
- RBDCoV-ACE2: 100% ACE2 binding inhibition indicates maximum binding inhibition between the corresponding SARS-CoV-2 RBD and ACE2. Responders are classified above an ACE2 binding threshold of 20%, as described in Junker et al. ¹

3 - There are several statements in results that should go in the methods section. Please, cite them in methods (lines 219? and certainly 222-224, among others that repeat methods)

Response: We appreciate your comment, but believe that in this particular context those statements are justified to strengthen the actual relevance, accuracy of our results and validity of our approach as frequently samples from other ethnicities than the assay was validated with can result in inaccurate seroprevalence estimates.

4 - When evaluating the correlations, how were managed samples with undetectable or below the lower limit of detection? From Fig 3 it is not clear whether those value outside the thresholds were/how used to the Spearman's correlation.

Response: We apologize if this is not clear in the figure legend. We included all samples in the correlation analysis. We have clarified this by expanding the figure legend to read: "*Spearman's correlation analysis was performed across all samples.*"

5 - When discussing seroprevalence ranges from other studies, please, state if they were population-based (sampling schemes) and use confidence intervals. This will help the readers.

Response: We have now added this information, if available in the cited manuscripts, in line 262 and line 265, respectively.

6 - This reviewer is not sure the authors have data to support statements as "immunity was reduced for both groups compared to European populations that we have previously studied". How comparable were the sampling? Was there a robust evidence of difference? Please, revise and justify.

7 - Following item 7, this statement in the abstract "in particular ACE2 binding inhibition were lower than expected within Colombian samples" is not clear to this reviewer.

Response 6&7: We discuss this finding in line 277-287 in the discussion. We base our observation on studies where the two multiplex-based SARS-CoV-2 immunoassays were also used to measure humoral immunity after infection. While we have included those studies as hyperlinks here for a direct review (Renk: <https://doi.org/10.1038/s41467-021-27595-9>; Junker: DOI: [10.1093/cid/ciac498](https://doi.org/10.1093/cid/ciac498) and DOI: [10.1038/s41598-022-10987-2](https://doi.org/10.1038/s41598-022-10987-2)), we also describe those studies from line 288 in more detail as suggested to read. "*In one study, sera samples from a comparable timeframe (2020-2021) were analysed from children exhibiting mild or asymptomatic infections as part of a household transmission study, while the other study included a broad sample selection from infected individuals (among them 20 paediatric samples with PCR confirmed infection from the first epidemic wave in Germany) as well as from vaccinated individuals.*" We also included additional publication for your reference here : Laub: <https://doi.org/10.3389/fped.2021.678937>; Yung: [doi:10.1001/jamapediatrics.2022.3072](https://doi.org/10.1001/jamapediatrics.2022.3072); Weinsberg: <https://doi.org/10.1038/s41598-020-00826-9>; Petrara: <https://doi.org/10.3389/fimmu.2021.741796> or Chiara: [doi:10.1001/jamanetworkopen.2022.21616](https://doi.org/10.1001/jamanetworkopen.2022.21616)). Although those reports use different assays to measure neutralization and antibody responses, they clearly demonstrate that children in general mount a robust titre and (correlating) neutralization response after SARS-CoV-2 infection which contrasts to our findings.

In addition, we have revised the last sentence in the abstract to read: “Overall, antibody titers, but in particular ACE2 binding inhibition were low within Colombian samples, requiring further investigation to determine any potential clinical significance.”

Overall, the manuscript would benefit from a text review assuring a narrative flow. The current format, particularly of results, is hard to follow.

Response: We acknowledge this comment and attempted to improve the narrative throughout.

Minor

1 - Could the authors please expand the sentence about "symptomatic infections are rare" (Lines 81/82). In which context? Compared with what? General, vulnerable population? Do we have accurate data for this statement?

Response: We have clarified the statement originating from from the first systematic on SARS-CoV-2 infection in children and adolescents in line 82 to read: “Previous studies on SARS-CoV-2 in children have identified that while most infections are associated with mild symptoms only², they are correlated with high viral loads³”.

2 - "unadjusted seroprevalence" (Line 98), in terms of test performance? sampling weights?

Response: We have now added a remark in line 99 to clarify the term unadjusted seroprevalence to read: “The same study estimated an unadjusted seroprevalence without considering sampling weights, test performance, or clustering level in the whole Colombian territory of 36% (95% CI: 34%–39%) for children aged 5-10 years, of 38% (95% CI: 36%–40%) for children aged 10-18 years, and of 51% (95% CI: 48%–55%) in indigenous populations.”

3 - Is this correct "Eight participants (47.1%)"? which denominator...

Response: Our percentage calculation was based on a total of 17 SARS-CoV-2 seropositive children in our urban study population. We have included the corresponding denominator (n=17) now for clarity from line 228 onwards.

4 - Please, use sex instead of gender, unless you evaluated gender.

Response: We completely agree and have now corrected this term in Table 1.

Reviewer #2

Remarks to the Author

The manuscript "Humoral immune response towards SARS-CoV-2 variants of concern and endemic coronaviruses in urban and indigenous children from Colombia" by Fernández Villalobos et al. presents data on several serological techniques to measure SARS-CoV-2 and common cold coronaviruses specific antibodies in children from Colombia belonging to both urban (n=80) and rural settings (n=82) at a single time point (March-April 2021). The study also adds a surrogate neutralizing assay to measure neutralization against different SARS-CoV-2 VoCs.

The study is technically solid, and it is well presented however the main problem is the limited relevance of the information on such a small number of individuals analyzed at just one specific moment.

Response: Thank you for your compliments. We are aware of our limitation in terms of sample size. However, the urban samples in our study were from a study that was recruiting participants during the first waves of the COVID-19 pandemic. This resulted in a low participation due to the multiple restrictions in place in Colombia. Our sample size is also comparable to most studies examining SARS-CoV-2 and hCoV immune responses in children (for instance: Shrwani: <https://doi.org/10.1093/infdis/jiab333>; Dhochak: <https://doi.org/10.1016/j.jcvp.2022.100061>; Thiriard: <https://doi.org/10.3389/fimmu.2023.1107156>) or to the few other studies which include analysis of neutralization responses after vaccination or infection in cohorts from South America. (Beltran-Paez: DOI: [10.1126/sciadv.abe6855](https://doi.org/10.1126/sciadv.abe6855); Acevedo: <https://doi.org/10.1038/s41564-022-01092-1> and <https://doi.org/10.1016/j.cmi.2022.11.017>; de Castro: <https://doi.org/10.1186/s12979-022-00310-y>, Alvarez-Diaz: DOI: <https://doi.org/10.3390/vaccines10020180> or Lopera: <https://doi.org/10.3389/fimmu.2022.879036>). The above cited studies from South America do not contain data from indigenous communities nor from children and most focus on vaccination responses making our study to the best of our knowledge the first study to compare those populations.

Furthermore, time of SARS-CoV-2 infection is lacking and taking into account the small number of participants significant biases can occur such as inclusion criteria, not completely define (how many offered?) or even exclusion as the authors describe in line 151: (from 99 indigenous children) we excluded 17 individuals with a SARS-CoV-2 positive PCR/antigen test result at the time of serum sampling from our analysis. This is a high rate of what seems to be active SARS-CoV-2 infection.

Response: We thank you for your remark. We purposely excluded positive PCR samples in indigenous population because these samples will not be comparable to the rest of the samples in terms of immune response as seroconversion has not even have occurred at time of sampling in fact introducing a well-defined bias in the population to be compared. For clarification and to accommodate your comment, we have now included two flow charts in the supplementary methods as Fig S1 and S2 to describe sample selection in more detail from the preexisting sample pool.

The rate of active and previous infections in the Wiwa community was already published as part of the previous study (<https://doi.org/10.3390/vaccines9101120>). We attributed those numbers to more crowded and disadvantaged living conditions and absence of social distancing in the Wiwa community as described already in the previous publication. Higher levels of infection/seroprevalence are however also a result of a systematic review performed for the first COVID-19 wave across the entire South Americas by Núñez-Zapata et al. (DOI:<https://doi.org/10.1016/j.ijid.2021.07.022>) or by Naeimi et al.2022(<https://doi.org/10.1016/j.eclinm.2022.101786>).

In this same line, the rest of the information, ie the lack of association between the levels of antibodies against SARS-CoV-2 and common cold coronaviruses along with the low level of neutralization observed is already well established or lacks adequate controls respectively. Overall, it is very difficult that the current design of study can provide any significant or informative conclusion.

Response: We want to restate that the present study is the first of its kind for children from Colombia that presents detailed data on SARS-CoV-2 antibody responses including neutralization capacity and humoral immunity towards the four common cold coronaviruses in urban and indigenous populations. Equally, we believe as stated in the discussion and observed in this studies, the low levels of neutralization in seropositive individuals is a novel finding as it contrast to findings of others studies (Renk: <https://doi.org/10.1038/s41467-021-27595-9>; Junker: DOI: [10.1093/cid/ciac498](https://doi.org/10.1093/cid/ciac498) and DOI: [10.1038/s41598-022-10987-2](https://doi.org/10.1038/s41598-022-10987-2)) where the two multiplex-based SARS-CoV-2 immunoassays were also used to measure humoral immunity after infection: We also included additional publications for your reference further below that demonstrate that children in general mount a robust titre and (correlating) neutralization response after SARS-CoV-2 infection which contrasts to our findings. (Laub: <https://doi.org/10.3389/fped.2021.678937>; Yung: [doi:10.1001/jamapediatrics.2022.3072](https://doi.org/10.1001/jamapediatrics.2022.3072); Weinsberg: <https://doi.org/10.1038/s41598-020-00826-9>; Petrara: <https://doi.org/10.3389/fimmu.2021.741796> or Chiara: [doi:10.1001/jamanetworkopen.2022.21616](https://doi.org/10.1001/jamanetworkopen.2022.21616)). We also want to refer to the “unbalanced” antibody levels between antigens that we observe in our indigenous children population where nucleocapsid-specific IgG is not induced or not present anymore, making this a critical point for the appropriate selection of analysis tools to provide accurate seroprevalence estimates as proxy for previous pathogen exposure.

Reviewers' comments:

Reviewer #1 (Remarks to the Author):

I have not further comments.

Reviewer #2 (Remarks to the Author):

The response does not include any information supporting the actual significance of the study apart from its descriptive nature

Helmholtz Centre for Infection Research GmbH | Department of Epidemiology
Inhoffenstraße 7 | 38124 Braunschweig | Germany

Editorial Board of Communications Medicine

Braunschweig, 02.06.2023

Dear Reviewer,

On behalf of all the authors, I would like to re-address the significance of our study
“Humoral immune response towards SARS-CoV-2 variants of concern and endemic coronaviruses in urban and indigenous children from Colombia”.

We acknowledge that our study is primarily descriptive in nature; nevertheless, it holds significant value in increasing our understanding of the impact of COVID-19 on diverse communities. By focusing on children from both urban and from indigenous populations, our research provides not only evidence on the transmission dynamics of the virus in densely populated urban areas as well as remote indigenous communities but also describes antibody responses in more detail than other studies resulting in the noteworthy finding of low neutralization and antibody levels.

This finding is particularly significant as it underscores the variability in humoral immune responses to SARS-CoV-2, suggesting the need to adjust serological tools when comparing different ethnic populations or differences in the immune response per se. Additionally, our studies examines humoral immunity towards the four common cold coronaviruses. These insights fill gaps in existing knowledge, especially in countries like Colombia, where such areas have been inadequately studied and can act as basis for future more mechanistic work.

By offering these insights, our study can serve as a valuable resource for informing public health interventions such as non-pharmaceutical interventions or vaccination strategies. Our findings can further guide efforts to address health inequalities and facilitate evidence-based decision-making to guide interventions strategies in the next pandemic.

On behalf of all authors, with kind regards

Monika Strengert

Dr. Monika Strengert
Department of Epidemiology
Tel. +49 531 6181-3103
Monika.Strengert@
helmholtz-hzi.de

**Helmholtz Centre for Infection
Research GmbH**

SCIENCE CAMPUS Braunschweig-Süd
Inhoffenstraße 7
38124 Braunschweig
Germany
Tel +49 531 6181-0
Fax +49 531 6181-2655
www.helmholtz-hzi.de

Chair of Supervisory Board:
MinDir'in Prof. Dr. Veronika von Messling
Federal Ministry of Education and
Research

Deputy:
MinDirig Rüdiger Eichel, Department
Leader, Ministry for Science and Culture
of Lower Saxony

Scientific Director:
Prof. Dr. Dr. h.c. Dirk Heinz

Administrative Director:
Christian Scherf

Registered Office:
Braunschweig

Registry Court:
Amtsgericht Braunschweig HRB 477
VAT Reg. No DE 11 48 15 244

Bank account:
Braunschweigische Landessparkasse
BIC NOLADE 2H XXX
IBAN DE56 2505 0000 0002 0588 81

St.-Nr. 13/200/24006